# Segmentation-Free Estimation of Left Ventricular Ejection Fraction Using 3D CNN Is Reliable and Improves as Multiple Cardiac MRI Cine Orientations Are Combined

**DOI:** 10.3390/biomedicines12102324

**Published:** 2024-10-12

**Authors:** Philippe Germain, Aissam Labani, Armine Vardazaryan, Nicolas Padoy, Catherine Roy, Soraya El Ghannudi

**Affiliations:** 1Department of Radiology, Nouvel Hopital Civil, University Hospital, 67091 Strasbourg, France; aissam.labani@chru-strasbourg.fr (A.L.); catherine.roy@chru-strasbourg.fr (C.R.); soraya.elghannudi-abdo@chru-strasbourg.fr (S.E.G.); 2ICube, University of Strasbourg, CNRS, 67000 Strasbourg, France; vardazaryan@unistra.fr (A.V.); npadoy@unistra.fr (N.P.); 3Department of Nuclear Medicine, Nouvel Hopital Civil, University Hospital, 67091 Strasbourg, France

**Keywords:** left ventricular ejection fraction, cardiac MRI, deep learning, convolutional neural network, inter-method discrepancy, multiple orientations combination

## Abstract

Objectives: We aimed to study classical, publicly available convolutional neural networks (3D-CNNs) using a combination of several cine-MR orientation planes for the estimation of left ventricular ejection fraction (LVEF) without contour tracing. Methods: Cine-MR examinations carried out on 1082 patients from our institution were analysed by comparing the LVEF provided by the CVI42 software (V5.9.3) with the estimation resulting from different 3D-CNN models and various combinations of long- and short-axis orientation planes. Results: The 3D-Resnet18 architecture appeared to be the most favourable, and the results gradually and significantly improved as several long-axis and short-axis planes were combined. Simply pasting multiple orientation views into composite frames increased performance. Optimal results were obtained by pasting two long-axis views and six short-axis views. The best configuration provided an R^2^ = 0.83, a mean absolute error (MAE) = 4.97, and a root mean square error (RMSE) = 6.29; the area under the ROC curve (AUC) for the classification of LVEF < 40% was 0.99, and for the classification of LVEF > 60%, the AUC was 0.97. Internal validation performed on 149 additional patients after model training provided very similar results (MAE 4.98). External validation carried out on 62 patients from another institution showed an MAE of 6.59. Our results in this area are among the most promising obtained to date using CNNs with cardiac magnetic resonance. Conclusion: (1) The use of traditional 3D-CNNs and a combination of multiple orientation planes is capable of estimating LVEF from cine-MRI data without segmenting ventricular contours, with a reliability similar to that of traditional methods. (2) Performance significantly improves as the number of orientation planes increases, providing a more complete view of the left ventricle.

## 1. Introduction

The left ventricular ejection fraction (LVEF) measure constitutes the cornerstone of cardiac evaluation because it summarises, in a single, easy-to-understand value, a major functional feature of the left ventricle. The LVEF has major prognostic significance, which greatly influences therapeutic conduct [1].

LVEF quantification relies on the measurement of left ventricular diastolic and systolic volumes (LVEF = (diastolic volume − systolic volume)/diastolic volume) and thus normally requires boundary tracing of endocardial contours, relying on a geometric model to quantify ventricular volumes (e.g., Simpson’s method). Echocardiography, iodinate or gamma-ventriculography, ECG-gated CT scan, or cardiac MRI are common methods used to quantify the LVEF.

The estimation of LVEF faces several challenges across different imaging modalities: variability between imaging methods (partially related to imaging quality dependence); volume underestimation with 2D echocardiography, although this has less impact on LVEF calculations [2]; and geometric assumptions. Even cardiac MRI, which is often considered the “gold standard” due to its high accuracy and reproducibility, has several issues, such as breath-hold requirements, which can be challenging for some patients and may lead to the misalignment of image slices, deleterious arrhythmia effects, basal slice selection, and papillary muscle and trabeculation inclusion.

Calculating LVEF by these different methods is a time-consuming process and prone to significant inter-method or inter-reader variability and discrepancies, which are related to the subjective nature of the measurements. This has been widely studied in the literature. For example, the inter-modality variability in LVEF measured by echocardiography, single-photon emission tomography (SPECT), and CMR was studied by Pellika et al. [3]. The correlations observed between these three methods were limited, with r in the 0.49–0.66 range, and Bland–Altman limits-of-agreement ranges (LARs) [4] were high, ranging from 28 to 35. Inter-observer and intra-observer variabilities for a given imaging method are generally lower.

Alternatively, alongside these explicit “analytical” methods that rely on the segmentation of ventricular contours, an implicit “synthetic” approach has also been considered, which is based on the visual estimation of LVEF by trained operators. Such visual (eyeballing) estimation of LVEF showed levels of imprecision similar to those observed between two quantitative methods. For example, a comparison between cine-MR quantification and visual assessment (n = 54) provided a LAR score of about 20 points [5].

The measurement of LVEF using artificial intelligence (AI) has also been undertaken with these two different approaches: explicit “analytical” with segmentation and implicit “synthetic” without contour tracing. The large majority of such work has been devoted to automatic segmentation methods (with contour tracing). Ouyang et al. [6] built a large, publicly available echocardiogram database, EchoNet-Dynamic, with which multiple studies have been carried out. A comparison between AI-based segmentation and reference methods provided an excellent mean absolute error (MAE) in the range of 4–5 points (using four-chamber-view echocardiography).

Fewer AI studies have focused on an implicit “synthetic” approach to LVEF that does not rely on the extraction of ventricular contours (i.e., no segmentation). This approach can be compared to visual estimation by a human operator and is attractive because it provides an instant snapshot of overall left ventricular function without any manual intervention. The Boston group’s landmark study [7] (n = 64,028), which remains a reference for automatic left ventricular ejection fraction (LVEF) determination, employed this approach rather than the segmentation method.

Two families of computer vision tools have been tested in this field: deep learning methods based on a convolutional neural network (CNN) [8,9,10,11,12] and Transformers, which were developed more recently and use self-attention modules [13,14,15,16,17,18].

However, existing AI models are elaborate and specific; moreover, most of the published studies have only considered a single imaging orientation (mainly ultrasound long-axis four-chamber view). Therefore, we undertook this work with the aim of (1) evaluating standard, publicly available 3D-CNN models and (2) seeking to improve diagnostic performance by combining multiple imaging planes with different orientation views, as is performed in visual analysis.

The key aspects of this article’s structure are as follows: In the “Methods” Section, we focus on data preparation, 3D-CNN models, and approaches to combining different orientation planes. The “Results” Section primarily addresses the combinations of orientation planes and the performance in internal and external validation. The limitations are reviewed in the “Discussion” Section, which also explores the anticipated perspectives for Transformer models and practical implementation considerations.

## 2. Materials and Methods

### 2.1. Study Population

Cardiac cine-MR sequences from 1082 patients, which were acquired between 2021 and 2023 in our institution, were retrieved from the picture archiving and communication system (PACS). This study was approved by the Institutional Clinical Research Department of our university hospital, and all datasets were obtained and deidentified, with waived patient consent in compliance with the rules of our institution.

An internal test group comprising 149 patients examined after model training completion was studied. Additionally, a set of 62 examinations conducted by an independent team at the Rhéna clinic in Strasbourg was collected to serve as an external validation group.

### 2.2. Cine-MR Acquisitions

All images were obtained using a Siemens 1.5 Tesla Sola scanner (Erlangen, Germany). Only SSFP cine sequences were analysed, with TE/TR in the range of 1.6/3.5 ms, a slice thickness of 8 mm (spacing between slices 10%), and a 32-element cardiac coil. Imaging planes, illustrated in Figure 1, were as follows: vertical long axis (VLA), 4-chamber view (4C), and stack of short-axis (SA) views, except for most of the patients (n = 142) with systemic sclerosis, for whom short-axis acquisition was replaced by acquisition from axial slices. Standard short-axis CVI42 (V5.9.3) software (AI-based for ventricular contour detection) was used for LV volumes and ejection fraction calculation (Circle Cardiovascular Imaging, Calgary AB, Canada).

### 2.3. Image Preparation

Long- and short-axis cine-MR Dicom images from the PACS were prepared with dedicated software (see Figure 2) that performed several functions:Image deidentification;The storage of image sequences with indexing in a Sqlite database;The association of labels, including LVEF, in the database;The classification of the orientation plane;The selection of a region of interest centred on the left ventricle with level scaling in this region;The selection of the systolic frame (with the smallest left ventricular cavity).

In order to speed up data gathering, tasks 4 and 5 were carried out with the help of two deep learning operators dedicated to the recognition of the orientation plane and the choice of a region of interest centred on the left ventricle (trained with a VGG model from the first 300 cine-MRI sequences processed in this study). Further details are provided in Appendix A.

The choices proposed by the system could be corrected manually, if necessary, before being recorded in the database. The selection of the systolic frame (with the smallest LV dimension) was carried out manually by scrolling through the series of images contained in each imaging sequence.

### 2.4. Deep Learning Process with 3D Convolutional Neural Network

Python scripts were developed in WSL2 (Windows Subsystem for Linux) with Ubuntu 22.4.4 and Python 3.11.8. Keras library 2.14.0 and TensorFlow 2.14.0 backend were used for CNN implementation.

In a first attempt, we tried to extract the spatial features of cine-MR images with a 2D-CNN, followed by a recurrent neural network model LSTM (LongShort-Term Memory), but the long calculation times and the poor results obtained led us towards a 3D-CNN solution.

The 3D-CNN models used here were developed from reference 2D models by Roman Solovyev et al. and are publicly available from the repository https://github.com/ZFTurbo/classification_models_3D (accessed on 1 September 2024). With these models, the authors demonstrated the efficient detection of stalled capillaries in 3D brain images of Alzheimer’s disease [19]. Moreover, this solution allowed a weight-transfer method that re-uses weights from 2D models pre-trained on ImageNet for the initialisation of 3D networks (as similarly described by Merino et al. [20]). Hyperparameters were manually tuned.

Model training was performed with 5-fold cross-validation. Attention was paid to ensuring the strict distribution of images of a given patient in either the training set or the validation set in accordance with CLAIM recommendations (item 21) [21].

In order to limit memory load, images were cropped to 128 × 128 frames and centred in the left ventricular region of interest, and the number of time steps was limited to 9 by interpolating the cine images from the 1st one up to the systolic frame defined in the preparation step (as described above). Data augmentation (using tf.data API, version TensorFlow 2.x) was applied to the 3D training set with rotation angles spanning from −25 to +25°, and the basic input shape was (n, 128,128,9). Due to the small number in the time dimension (9), the stride values of the CNN layers were changed to (2,2,1).

The final fully connected layer, after using GlobalAveragePooling3D (more favourable solution than Flatten in our experiment), used Dense (1) and activation “linear” for the regression task. Hyperparameters were set as follows: 140 epochs, batch size 4–16 (depending on the risk of memory overflow), and Adam optimiser with a learning rate of 10^−4^. The loss function used during fitting was MAE (mean absolute error between labels and predictions), and for each of the five folds, the model corresponding to the best loss was retained using a Keras checkpoint callback.

Firstly, several 3D models were tested only on long-axis views in order to select the best option for the comparison of different combinations of orientation planes. Then, several combinations of orientation views were trialled in order to find the best method for estimating the LVEF. Figure 3 illustrates the different approaches that were used: (1) the simple pooling of long-axis or short-axis views, (2) the pasting of long-axis or/and short-axis frames into a composite panel (single model), and (3) the concatenation of two specific models corresponding to the pasted long- and short-axis composite panels. Short-axis levels were centred in the middle of the left ventricle.

### 2.5. Statistical Analysis

The results are expressed in the form of different metrics to be comparable with most of the previously published studies. We calculated the following metrics: Pearson correlation coefficient r; MAE: mean absolute error = (1/n) × ∑|Predicted*_i_* − Actual*_i_*|, where Predicted*_i_* is the calculated value, Actual*_i_* is the actual value, and n is the number of observations; RMSE: root mean square error = sqrt((1/n) × ∑(Predicted*_i_* − Actual*_i_*)^2^); LAR (95% Bland and Altman limits-of-agreement range [4]); and coverage percentage (the proportion of observations that fall within 5% of the paired absolute difference [3]). Sensitivity, specificity, and area under the ROC curve (AUC) for LVEF < 40% and for LVEF > 60% were computed.

Mean and standard deviation values for all of these metrics were determined across the 5 folds, as this adheres to the cross-validation principle and provides a better summary of model performance and stability on unseen data. Additionally, in order to visualise the overall predictive performance on the complete dataset, a global distribution diagram that aggregates actual LVEF and predicted values across all 5 folds of the cross-validation was drawn, and global metrics on these aggregated data were calculated.

The Shapiro–Wilk test was used to check the normality of distributions. Median values and interquartile ranges (IQRs) are reported in cases of skewed distributions. Two-sample Student’s *t*-test with pooled variance was used to compare results obtained with different combinations of orientation planes.

## 3. Results

### 3.1. Population Profile

The study population included 508 women (47%), showed a mean age of 55.7 ± 16.8 years, and showed a mean weight of 77.5 ± 19.9 kg. The following MRI diagnoses were made in the study population: dilated cardiomyopathy (n = 115), hypertrophic cardiomyopathy or left ventricular (LV) hypertrophy (n = 139), ischaemic heart disease (n = 102), valvular heart disease (n = 31), cardiac amyloidosis (n = 45), systemic sclerosis (SSc: n = 190), sarcoidosis (n = 12), pericardial effusion (n = 20), Fabry disease (n = 21), myocarditis (n = 43), and other diagnoses (n = 41). MR findings were normal in 323 patients referred for other reasons.

The LVEF distribution was not normal: median 65 and IQR 18 (mean 61.6 ± 15.0). The mean LV mass was 143.0 ± 55.0 g, the mean LV volume was 153.2 ± 63.2 mL, and the mean systolic phase was 9.8 ± 1.3. In the group of patients with short-axis slices (n = 910), the mean number of short-axis slices in the stack was 6.7 ± 1.8. The corresponding histograms are displayed in Figure 4.

### 3.2. Screening for Best CNN Model

Table 1 summarises the comparison of several models trained only on long-axis views in order to select the best model for further experiments. The performance appeared to differ little between the tested models. We selected 3D-Resnet18, as it provided the best results and had the shortest calculation time with a limited number of parameters.

The role of ImageNet weights was very limited compared to random value initialisation, perhaps due to the specific nature of cine-MR images, but ImageNet weights were always used in our experiments.

### 3.3. Results According to Various Combinations of Imaging Planes

Table 2 shows the results obtained with the 3D-Resnet model using various combinations of imaging planes. The number of samples, n, varies depending on the combination of the inputs used, because all six SA views were not available for all patients. Relying on long- or short-axis views alone (as in the two first lines of the table) provides the weakest results; the combination of several long-axis (2LA) or several short-axis (2SA) views in composite panels improved all metrics significantly. The combination of complementary orientation planes (2LA + nSA, up to n = 6 short-axis slices) in composite panels provided the best results. The more short-axis slices there were, the better the results were. The concatenation of separate models for multiple long- and short-axis views did not consistently provide better results than a single model of a composite panel (last lines of Table 2).

We did not seek to increase the number of short-axis slices beyond six because some patients did not have useful additional slices and because extreme levels, at the base or at the apex, resulted in the loss of the LV cavity in the systole.

Within the series of experiments performed, the sensitivity, specificity, and AUC values followed the same trend as the other metrics listed in Table 2 (i.e., optimal results were obtained by combining the maximum number of slices). For the best configuration identified (2LA + 6SA composite panel), we found a sensitivity of 0.90, a specificity of 0.95, and an AUC of 0.99 for LVEF < 40% and 0.95, 0.80, and 0.97, respectively, for LVEF > 60%. Figure 5 shows the correlation and Bland–Altman diagrams from the best experiment performed (using a composite panel of two long-axis and six short-axis slices).

As shown in Figure 5, poor results were seen in some patients, with discrepancies between the actual and predicted LVEF values exceeding ±15%. An examination of these unfavourable cases showed either that the model was malfunctioning (approximately half of all cases) or that there were artefacts in the images or labelling errors attributable to an imperfect or questionable drawing of LV contours with the CVI42 software. Figure 6 illustrates such examples.

### 3.4. Validation Tests

An internal validation test carried out on 149 additional observations including 2LA and 6SA orientation views (not normally distributed: median 64, IQR 9) provided good results, very close to those obtained with the best model in the 2LA + 6SA experiment. Inference on these 149 previously unseen patients provided the following results: LAR = 23.59, MAE = 4.98, RMSE = 6.34, r = 0.83, and CP = 55.70% (see Appendix A).

A decrease in the performance of the best model was observed in the external validation group of 62 patients, for which data were normally distributed but with a lower median = 55 instead of 64–65 for the previous groups (mean 53.30% ± 15.96%): LAR = 30.93, MAE = 6.59, RMSE = 7.76, r = 0.89, and CP = 41.94% (see Appendix A).

## 4. Discussion

### 4.1. Main Contributions of the Study

This study shows that the use of standard 3D-CNN models makes it possible to estimate the LVEF with excellent reliability without the need for specific or complex models, as used in most similar studies. Our algorithm performs similarly to visual analysis performed by an experienced cardiologist and provides good agreement with the quantitative LVEF obtained with LV contour segmentation, for all evaluated metrics, on a dataset of more than 1000 cine-MR exams from our institution. As summarised in Table 3, our results are very close to the best results published from studies using the usual segmentation methods and specific machine learning/Transformer architectures without segmentation. For example, the MAE index obtained in our study was 4.97, whereas the optimal values reported in the literature with ultrasound and more sophisticated techniques were around 4.0 (Table 3). AUC values for the diagnosis of severe cardiac dysfunction (LVEF < 40) and for the diagnosis of satisfactory cardiac function (LVEF > 60%) exceed 95%, which testifies to the reliable nature of our algorithm.

The second contribution of this study is that it shows the significant benefit provided by the combination of multiple orientation planes: assessing more than one view of the heart significantly improved the assessment of LVEF. In our work, two approaches to combining long-axis and short-axis orientation planes led to similar improvements in LVEF prediction: (1) the simple pasting of distinct orientation planes into a composite panel (processed with a single model) and (2) the use of two specific models applied to separate composite panels (long- and short-axis views). With these two methods, we observed that the more slices we combined, the more the LVEF prediction was improved (the gain observed was around 17% if we consider MAE and RMSE indices). This finding is consistent with the clinician’s experience that both long-axis views and a short-axis stack are necessary for a comprehensive assessment of cardiac function. This can be compared to the well-known improvement in the detection of many diseases (e.g., breast cancer, brain tumour, lung nodule) through the combination of multiple viewing angles. Although there is no mathematical theory to demonstrate this notion, several theoretical concepts may support this approach, such as providing complementary information, leading to “ensemble learning” at the input level, increasing the feature space (richness) available to the CNN, etc.

Table 3 indicates that few previous articles have analysed the advantage provided by the association of multiple orientation planes compared to a single imaging plane, as most of the research focused solely on the apical four-chamber imaging plane using ultrasound. Alven et al. [16] demonstrated that the concatenation of features resulting from several long-axis views with ultrasound improves prediction: “using all possible view instances boosts the model’s performance as it increases the amount of available training data but also increased computational complexity”. Based on CMR, Inomata et al. [11] combined four mid-ventricular short-axis planes but did not compare these results with those from long-axis planes. In keeping with our findings, in the CMR study of Liu et al. [10], an MSE value of 7.32 obtained with three slices improved to 7.11 with five slices and to 6.84 with seven short-axis slices, leading the author to state that “a large number of slices preserves more information and the 3D structure is a more complete input to the model”. However, no other study to date has combined long-axis and short-axis orientations as we did.

### 4.2. Transformers as an Alternative to CNN

Aside from CNNs (used in this work), the Vision Transformer (ViT) introduced by Dosovitskiy et al. in 2020 [22] constitutes a breakthrough in the field of computer vision [23]. Images are divided into patches (tokens) organised into a sequence, which is then fed into an encoder. In the encoder, the self-attention mechanism enables the model to determine the relative importance among tokens [24]. This allows the model to understand the context. Several families of ViTs have been proposed (DeiT, BeiT, Swin, hybrid…). The implementation of these models is, however, significantly more challenging than that of CNNs, especially for 3D data. However, the Hugging Face environment provides powerful tools for implementing various types of Transformers (preferably with PyTorch rather than Python, e.g., [14]).

Table 3 shows that Transformers and BERT (Bidirectional Encoder Representations from Transformers) [13,14,15,16,17,18] generally outperform CNNs [9,10,11,12,25,26,27,28] for this task. This was also shown by estimating the LVEF from the movement of the coronary arteries based on coronary angiography [29]. Transformers offer a superior capability to incorporate temporal information compared to 3D-CNN. Alven et al. [16] noticed that having BERT as a terminal layer was crucial for optimal performance. To date, Transformers have not been used on cine-MR datasets. However, it should be noted that the implementation of Transformer architectures (such as Swin Transformer/K-Net or Segformer) for endocardial contouring may also be used for segmentation (Liao et al. [30]), allowing improved delineation of the boundary between the left ventricle and mitral valve, thereby preventing the erroneous segmentation of the left atrium and left ventricle as a single unit.

**Table 3 biomedicines-12-02324-t003:** Results from the literature addressing discrepancies between LVEF obtained by several echocardiographic and CMR methods.

	Authors	Dataset	View Slices	Model	r or R^2^	MAE	RMSE	LAR	Classification
**ECHO SEGMENTATION**	Ouyang 2020 [6]	Int EcNet (1288) External (2895)	4C (1)	EchoNet Dynamic	R^2^ 0.81 R^2^ 0.77	4.1 6.0	5.32 7.7		50% → AUC 0.97 50% → AUC 0.96
	Tromp 2022 [31]	Int (142) Ext Canad (748) Ext Taiw (7724) Ext EcNet (6286)	LA	UNet	r 0.89 r 0.75 r 0.75 r 0.76	5.5 8.6 10.2 6.5	6.8 10.8 12.6		40% → AUC 0.96 40% → AUC 0.91 40% → AUC 0.90 40% → AUC 0.92
	Leclerc 2019 [32]	Val Camus	4C (1)	UNet++	r 0.80	5.6			
	Vega 2024 [33]	Val (441)	PLAX 1	ExoAI com	r 0.89		7.29		50% → accur 91%
	Zhang 2024 [34]	Int EcNet (1270) Ext Camus (200)	4C (1)	DL model EchoNet	r 0.83 r 0.79				40% → AUC 0.98 40% → AUC 0.90
	Moal 2022 [35]	Int (200) Ext (450)	4C/2C (2)	DL model n = 783		6.1 5.39			
	Batool 2023 [36]	Int EcNet (1277)	4C (1)	DL LSTM	r 0.78	5.73	7.72	30.2	
	Sveric 2023 [37]	Ext (889)	LA (2)	LVivoSeamles	r 0.90			26.1	
**ECHO REGRESSION**	Blaivas 2022 [9]	Val EcNet	4C (1)	VGG-LSTM	r 0.35	8.1	12.0	34.3	
	Akan 2024 [18]	Int EcNet (1200)	4C (1)	ViViT	R^2^ 0.55	6.14	8.4		50% → AUC 0.83
	Reyna. 2021 [13]	Val EcNet	4C (1)	Transform.	R^2^ 0.52	5.95	8.38		
	Muhta. 2022 [14]	Val EcNet	4C (1)	Uniformer	R^2^ 0.82	3.95			
	Fazry 2023 [15]	Val EcNet	4C (1)	Swin Transf.	R^2^ 0.59	5.6	7.59		
	Alven 2024 [16]	Val EcNet Val Camus	LA (4)	3DCNN + BERT	R^2^ 0.84	4.0		20.8	
	Batool 2024 [17]	Val EcNet	4C (1)	EFNet hybr	r 0.88	4.35	5.83		
	Lau 2023 [7]	Ext C3PO (9256) Ext EcNet (10,030)	LA (several)	DROID Train 64028	R^2^ 0.74 R^2^ 0.69	4.23 5.23		21.0 25.5	
**MRI SEGMENT**	Assadi 2024 [25]	Val 3 sets (814) Ext (101)		U-NET	r 0.91			20.6	
	Hatipoglu 2022 [26]	Ext (300) Ext (20)	SA stack	CVI V5.12	r 0.95			18.3	
	Backh. 2019 [27]	Ext (300)		suiteHEART	r 0.95	SD 5.9		(26.0)	
	Mariscal 2023 [28]	Int (414) Ext (6888)	SA stack	nnU-Net	r 0.86	3.4 * 5.3 *		21.04	* median, not mean absolute error
**MRI REGRESSION**	Liu 2020 [10]	Val (1140)	SA (7)	Densenet		6.84	9.74		
	Inoma. 2023 [11]	Val (100)	SA (4)	3DResNet50	r 0.80	9.41	12.2	39.0	
	Gheorghita 2022 [12]	Int Kaggle (440) Int UKBB (412)	SA (6)	CNN-LSTM	r 0.78 r 0.81	4.9 2.8	7.1 3.7	26.0 14.0	
	Our results	Val (910) Int (149) Ext (62)	LA + SA (2–8)	3D-Resnet18	R^2^ 0.83 R^2^ 0.69 R^2^ 0.79	4.97 4.98 6.59	6.29 6.34 7.76	25.0 23.6 30.9	40% → AUC 0.99

Segmentation is for an explicit method with contour tracing (grey background), and regression is for an implicit approach without segmentation (green and blue background). Datasets list the type and name of the cohort: Val: model development (training/validation); Int: internal validation; Ext: external validation (red color), along with the number of observations involved. View/slice: LA, long-axis view; 4C, 4-chamber view; SA, short-axis view, along with the number of slices entered into the model. MAE, mean absolute error; RMSE, root mean square error; LAR, 95% limit-of-agreement range. AUC, area under the ROC curve for a given LVEF cutoff. The Echonet dynamics (EcNet) database contains 10030 4-chamber echo videos. The Camus database contains 500 long-axis-echo videos.

### 4.3. Limitations

Several limitations of this study should be highlighted. Despite good results obtained on an internal validation test involving 149 patients, the lack of generalisation of our results constitutes the main shortcoming of the present work (MAE deteriorated from 4.98 with internal validation to 6.59 with external validation). Such a decrease in performance is not uncommon when models are applied to external datasets [6,28,31,34]). For example, in the study by Ouyang et al., MAE increased from 4.05 with an internal dataset to 6.0 with an external dataset [6]. Several differences between groups may explain these weaker results. For instance, the median LVEF was substantially higher in our skewed, main and internal validation datasets (median = 65% and 64%, respectively) compared to the external dataset from the Rhéna clinic, which was normally distributed with a median of 55%. As noted by Akan et al. [18] and Gheorghita et al. [12], this imbalance can adversely affect the performance of models trained on such skewed data, potentially leading to bias towards predicting values within the most prevalent range. Additionally, MR scanners, post-processing software, and human operators differed between the two groups. Furthermore, since the best model was selected based on a fixed number of epochs, it may have been overfitted to the training data, potentially leading to poorer performance on external datasets.

Quality control was not systematically performed on the whole dataset (in contrast to Mariscal-Harana et al., for example [28]), and some cases with image artefacts or erroneous ground-truth segmentation (particularly in the challenging short-axis basal slice, as shown by Retson et al. [38]) were not corrected or excluded, as was carried out in other studies [28,39].

The assumption that the diastolic phase corresponds to the first image of the cycle is not always accurate, which can skew the estimation of the diastolic volume and, therefore, the LVEF (on the other hand, the adaptation of the systolic phase to each cine sequence during our pre-treatment process was an advantage).

The lack of quantification of left ventricular volumes constitutes another limitation, as we focused on LVEF alone. However, it is also possible to carry out an implicit estimation of diastolic and systolic volumes, as for the LVEF [12], provided that the pixel size is normalised (which was not performed here). Furthermore, this task would not require the use of a series of cine images; the estimation performed on the diastolic and systolic images alone would be sufficient. It should also be noted that using synthetic data generated for pre-training a CNN (GauGAN) significantly improves the prediction [12].

Our approach does not adequately characterise potential anomalies in segmental contraction. Consequently, a thorough analysis of cardiac function should not be confined solely to the assessment of LVEF. Rather, it must incorporate regional segmentation to provide a more complete and nuanced evaluation of cardiac performance.

Finally, an inevitable criticism that is intrinsic to our implicit approach using a CNN rather than segmentation concerns the poor explainability of our predictions, in contrast to segmentation methods. Saliency Grad-cam heatmaps, which identify the main pixels responsible for the prediction, should be used but are not easy to implement or to analyse with a 3D set of data.

### 4.4. Position of This Work within the Framework of Automated LVEF Determination

As illustrated in Table 3, this study focuses on a limited subset of research concerning the automatic determination of LVEF. Considering the subject more generally, explicit segmentation methods are predominantly employed [32,33,36,37] due to their significant advantage of being inherently explainable, as the ventricular contour proposed by the model can be superimposed on the reference contour delineated by human experts. This explainability is crucial for the clinical acceptance of a diagnostic method. U-Net-type algorithms form the foundation of the models utilised in this context (e.g., Ono [40]).

Table 3 summarises several studies of this nature that employ metrics identical to those in our research. Among these studies, it is noteworthy to highlight the substantial sample sizes frequently employed for model training or testing. The Echonet cohort utilised 10,030 participants [6], while Lau et al.’s C3P0 cohort included 9256 subjects [7]. Approximately 5000 participants were drawn from the UK Biobank registry, which was notably used for Circle CVI42. Tromp’s study [31] employed a Taiwanese cohort of 7724 individuals. Five external datasets, including 6888 patients, served as an external group in the study by Mariscal-Harana et al. [28]. However, it is important to note that the evaluation of models on an external test group was not consistently performed across all studies (refer to Table 3 for details).

In addition to demonstrating good accuracy [31], Tromp et al. also emphasised the superior reproducibility of AI methods compared to human operators [41]. Similarly, other researchers, such as Ouyang et al. [6] and Olaisen et al. [42], have shown that automatic segmentation methods lead to improved test–retest reproducibility compared to inter- and intra-observer analyses.

An additional advantage of AI in determining the LVEF is its potential for use by novice operators in point-of-care ultrasound applications, particularly in emergency settings [43].

For MRI, several cardiac magnetic resonance (CMR) analysis software packages are now available (mentioned by Assadi et al. [25]), resulting in significant time savings for clinicians during post-processing [26]. Unlike ultrasound, segmentation is generally performed on a stack of short-axis slices while paying particular attention to the position of the mitral valve annulus in order to be able to correctly separate the left atrium from the left ventricle in systole (explicitly identified in Siemens’ Syngo Via software, for example). Segmentation (U-Net-based) on long-axis sections alone tends to underestimate ventricular volumes and overestimate LVEF but leads to as good a prediction of the prognosis as measurement on short-axis sections [25].

### 4.5. Future Work and Perspectives

Firstly, a future goal arising from this study will be to extend the present work to Transformer algorithms, which should improve performance. The good results observed already reinforce the interest in the deployment of this method for clinical implementation in parallel with the traditional quantification of LVEF by segmentation. The necessary processing pipeline would first require reliable automatic detection of the orientation plane, which is easily accomplished by a CNN, as reported by several studies (e.g., Vergani et al. [39]). Among the stack of short-axis slices, a discrimination must be made between the “useful” slices, which pass into the left ventricle, and the inappropriate slices to be rejected at the apex and base. Secondly, the identification of ventricular regions of interest for cropping and scaling proved to work well with CNNs, as was observed in the “image preparation” step of our study. Automatic identification of the end-systole (not performed here) is technically more difficult but has been described in several papers [6,17,34,44] and is not necessary if we use Transformers rather than CNNs.

Finally, it is important to note that although it is certainly important to seek to minimise gross errors corresponding, for example, to differences of ±10% between the reference LVEF and the prediction by the algorithm, it would be illusory to seek to further reduce small deviations. Indeed, there will always remain a variability that is intrinsic to this type of inter-method comparison and is linked to differences in the principles behind the measurement processes. In clinical practice, deviations of ±5% hardly change therapeutic behaviour; knowing that, in some cases, we are close to the 35% threshold at which the Madit 2 study recommends the implantation of an automatic defibrillator, human arbitration taking into account the entire patient file must always remain a priority.

## 5. Conclusions

Applying a 3D-CNN for the task of LVEF prediction without segmentation, through direct regression, performed similarly to visual analysis carried out by an experienced cardiologist, allowing the estimation of LVEF with excellent reliability without the need for specific or complex models. The combination of multiple orientation planes (two long-axis and six short-axis slices) led to the best results, which are very close to those published for the usual segmentation methods and specific machine learning/Transformer architectures without segmentation. It is possible to adapt this approach to be operational in current imaging devices; however, further model refinement is needed to improve generalizability. Specifically, the model should be tested and adapted across different patient populations and diverse clinical settings to enhance its robustness and applicability.

## Figures and Tables

**Figure 1 biomedicines-12-02324-f001:**
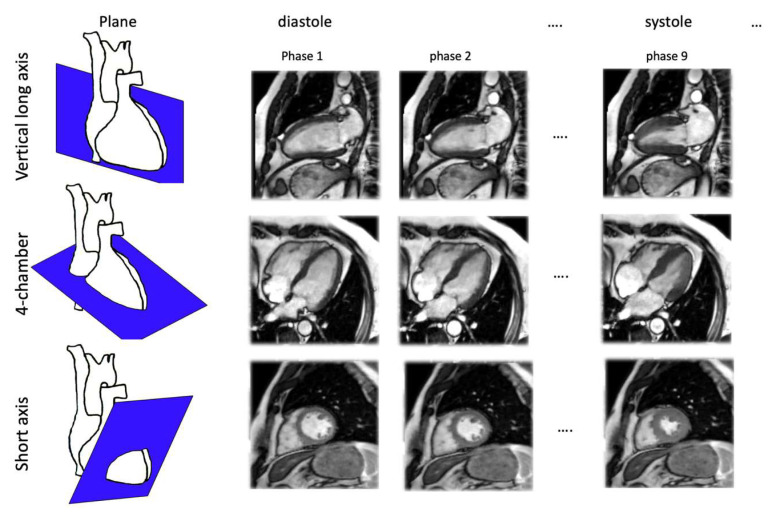
Sketch map of cine-MR imaging plane orientations: vertical long axis, 4-chamber view, and short-axis plane perpendicular to long-axis views (series of slices encompassing the left ventricle).

**Figure 2 biomedicines-12-02324-f002:**
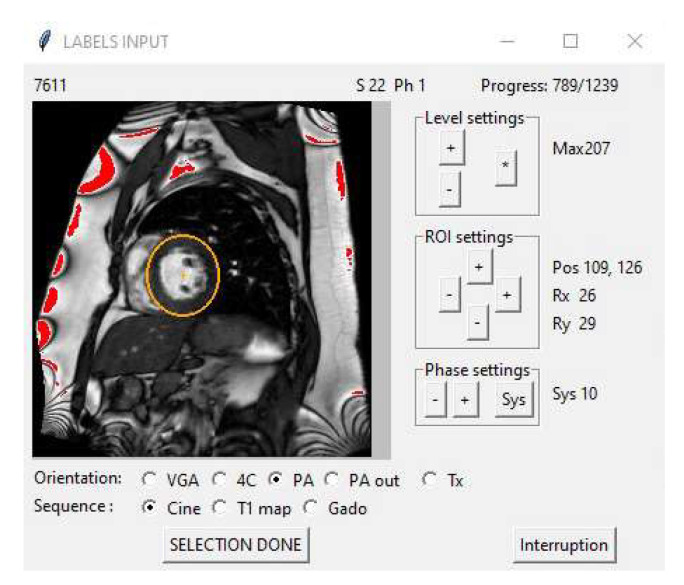
Preparation of Dicom images, including label recovery, automatic selection of the orientation plane, automatic centring of the ventricular region of interest (orange ellipse) with level scaling, and manual selection of the systolic phase.

**Figure 3 biomedicines-12-02324-f003:**
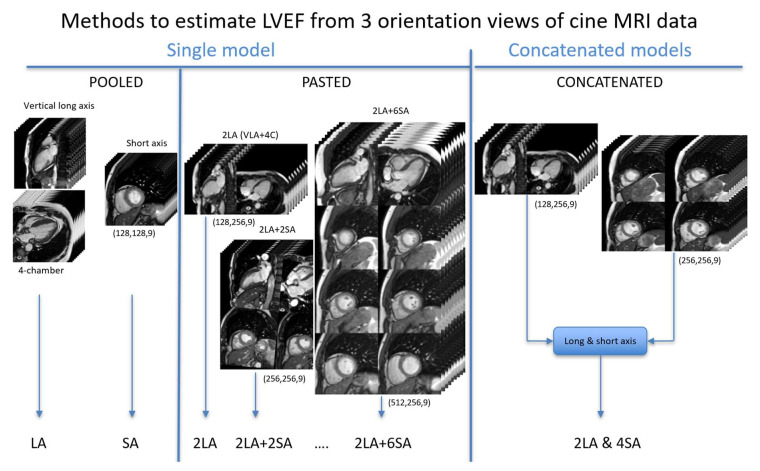
An illustration of different combinations of orientation planes and models used to estimate LVEF: “Pooled” from separate long- and short-axis frames; “Pasted” with panels of composite long-axis and/or short-axis frames (single model); and “Concatenated”, relying on two models based on composite long- and short-axis panels. Input shapes are indicated in parentheses.

**Figure 4 biomedicines-12-02324-f004:**
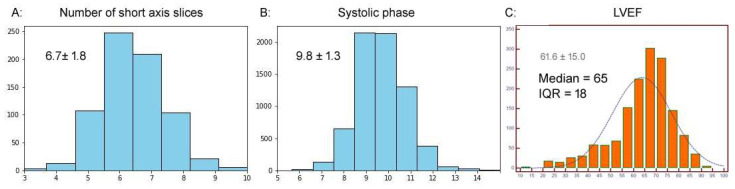
Histograms showing the distributions of the number of short-axis slices, systolic phases, and LVEF (the last not being normally distributed).

**Figure 5 biomedicines-12-02324-f005:**
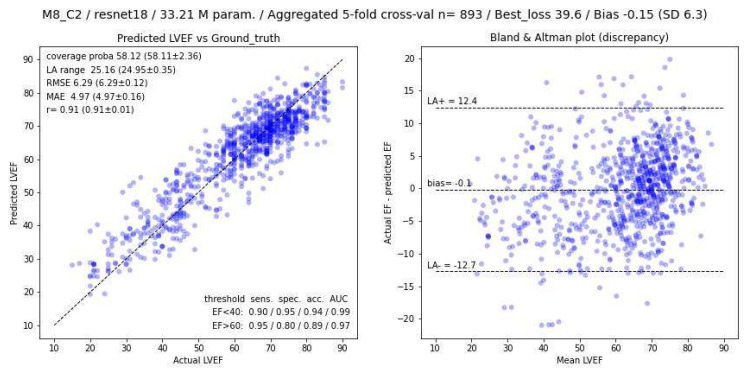
Correlation graph and Bland–Altman plot of inter-method discrepancies with corresponding 95% limits of agreements, indicated by horizontal dotted lines (LA+, LA−), using composite panels made from 2 long-axis slices and 6 short-axis slices.

**Figure 6 biomedicines-12-02324-f006:**
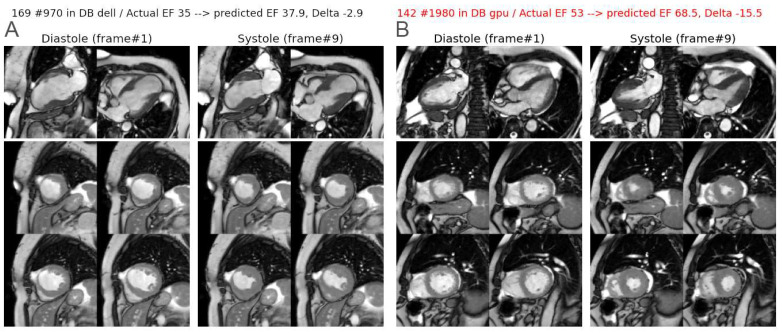
Two examples of LVEF prediction from composite panels containing 2 long-axis and 4 short-axis slices (only diastolic and systolic phases are displayed here). (**A**) Correct LVEF prediction in a case of antero-septo-apical aneurysm. (**B**) An example of incorrect prediction (text in red color) with the overestimation of LVEF by the algorithm, probably accentuated by concomitant underestimation by reference quantification.

**Table 1 biomedicines-12-02324-t001:** Five-fold CV results obtained with different 3D-CNN models for LVEF regression task.

Model	Layers	Parameters	Duration	LAR	RMSE	MAE	r
VGG19	22	60.1	33″	31.2	7.8	6.1	0.82
InceptionV3	311	34.1	31″	31.6	7.9	6.0	0.81
Densenet121	427	11.4	32″	30.5	7.6	5.9	0.83
Densenet201	707	25.6	33″	30.4	7.6	5.9	0.83
**Resnet18**	**86**	**33.2**	**31″**	**30.0**	**7.5**	**5.8**	**0.83**
Resnet50	190	46.2	37″	30.0	7.5	5.8	0.83
Resnet101	377	85.3	36″	30.8	7.8	6.1	0.82
Seresnet18	142	33.3	38″	31.6	7.9	6.2	0.81

Layers, number of layers in the model; Parameters, number of parameters of the model (millions); Duration, duration per epoch (seconds) for 1687 128 × 128 × 9 long-axis-view sequences. Metrics (LAR, RMSE, MAE, and r) were calculated from the best loss models obtained through 100 epochs (see text for acronyms).

**Table 2 biomedicines-12-02324-t002:** Five-fold CV results obtained with 3D-Resnet model for LVEF regression task, according to orientation plane combination.

	Combination	n	LAR	MAE	RMSE	r	CP	AUC40	AUC60
Pooled	LA (VLA,4C)	2133	29.91 ± 0.85	5.96 ± 0.12	7.48 ± 0.20	0.87 ± 0.01	49.81 ± 1.15	0.95	0.92
	SA (4)	3442	30.10 ± 1.60 *	5.85 ± 0.27 *	7.56 ± 0.38 *	0.87 ± 0.01	52.64 ± 3.12	0.97	0.94
Pasted	2LA (VLA + 4C)	1082	27.83 ± 1.23 *	5.50 ± 0.28 *	6.99 ± 0.32 *	0.89 ± 0.01	55.49 ± 3.48	0.96	0.95
	2SA	910	27.78 ± 1.69 ns	5.50 ± 0.21 ns	7.00 ± 0.37 ns	0.89 ± 0.02	52.52 ± 2.24	0.98	0.96
	2LA + 2SA	910	26.65 ± 1.36 *	5.28 ± 0.30 *	6.76 ± 0.37 *	0.90 ± 0.01	55.76 ± 1.42	0.97	0.95
	2LA + 4SA	910	26.39 ± 1.24 *	5.32 ± 0.27	6.65 ± 0.31 *	0.91 ± 0.01	53.63 ± 3.15	0.98	0.94
	**2LA + 6SA**	**893**	**24.95 ± 0.35 ** *****	**4.97 ± 0.16 ** *****	**6.29 ± 0.12 ** *****	**0.91 ± 0.01**	**58.11 ± 2.26**	**0.99**	**0.97**
Concatenated	2LA + 2SA	910	25.64 ± 0.59 *	5.18 ± 0.14 *	6.42 ± 0.15 *	0.91 ± 0.01	55.20 ± 2.11	0.98	0.94
	2LA + 4SA	910	26.31 ± 1.12 *	5.27 ± 0.16 *	6.65 ± 0.24 *	0.91 ± 0.01	56.16 ± 2.12	0.97	0.95
	2LA + 6SA	893	25.14 ± 0.87 *	5.12 ± 0.16 *	6.34 ± 0.15 *	0.91 ± 0.01	55.76 ± 2.22	0.99	0.97

Pooled-view results are shown in the 2 first lines. LA corresponds to long-axis (vertical long-axis and 4-chamber), and SA (4) corresponds to 4 central short-axis views. The results of five combinations of pasted frames are in the central part of the table. Pasting 2LA + 6SA slices led to the best results (bold). Concatenated model results are displayed in the bottom 3 lines. Metrics (LAR, RMSE, MAE, r, and CP) were calculated from the best loss models obtained through 140 epochs (see text for acronyms). AUC40 and AUC60 represent areas under the ROC curve for LVEF < 40% and for LVEF > 60%. Comparison with *t*-test: * means *p* < 0.001 when compared with the line above. ns means not significant.

## Data Availability

The central part of this work is based on the 3D-CNN network code, which is publicly available from the repository https://github.com/ZFTurbo/classification_models_3D (accessed on 11 October 2024). The entire code is available from the authors on reasonable request. Dicom images of the dataset cannot be made publicly available due to restricted access under hospital ethics.

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
