# Peer review of "Segmentation-Free Estimation of Left Ventricular Ejection Fraction Using 3D CNN Is Reliable and Improves as Multiple Cardiac MRI Cine Orientations Are Combined"

_biomedicines, 2024, doi:10.3390/biomedicines12102324_

Round 1
Reviewer 1 Report
Comments and Suggestions for Authors
Line 96 to 104==> I would prefer if this is in the results section rather than the methods since you are analyzing the profile of your population.
Same here: Mean LVEF was 61.6±15.0, mean LV mass was 143.0±55.0 g, mean LV volume was 141 153.2±63.2 ml, and mean systolic phase was 9.8±1.3. In the group of patients with short- 142 axis slices (n=910), the mean number of short-axis slices in the stack was 6.7±1.8. Corre- 143 sponding histograms are displayed in Figure 3.==> Needs to be moved to the results section in my opinion.
STatistical analysis: r (but a good correlation does not necessarily imply good agree- 199 ment), ==> This sentence does not belong here. If you want to metion that this would be a limitation in the discussion but not in the methods.
LVEF seems not normally distributed. I guess it would be better to report median and IQR.
IN terms of statistical analysis, since youa re creating a predictive tool I think the best metric to report would be R squared rather than r.
Overall great work!
Reviewer 2 Report
Comments and Suggestions for Authors
Dear Authors,
I would like to congratulate you on the hard work and the submission of an interesting and well-written manuscript. While I acknowledge the validity of your study, I have a minor suggestion. In certain cases, such as those involving complex left ventricular abnormalities, segmentation could provide additional insights that a segmentation-free model might overlook. Therefore, while your approach seems effective for general cases, incorporating an optional segmentation model could enhance its versatility. I believe this point is worth addressing in the discussion section.
Reviewer 3 Report
Comments and Suggestions for Authors
The paper presents a segmentation-free approach for estimating LVEF using deep learning. Overall, the work is well-organized, and the results look promising. However, there are a few key areas where improvements could be made.
First, while the introduction provides a solid background, it could benefit from a broader discussion of related AI models(e.g., transformers), to give a better sense of how this work fits into the larger landscape of deep learning applications in medical imaging.
One of the main limitations is the lack of external validation, which is acknowledged in the discussion. This makes it harder to generalize the findings to other datasets and weakens the claim of broad applicability. Although the paper compares its results with other similar approaches/related works, those methods aren’t fully evaluated on the same dataset, which makes it difficult to draw direct comparisons.
Another issue is the interpretability of CNN models. Currently, the paper doesn’t address how clinicians can understand the model’s predictions, which is crucial in medical applications. Using saliency maps or heatmaps to visually explain which features influence the model’s decisions could make the tool more useful and trustworthy for clinical practice.
Author Response
First, while the introduction provides a solid background, it could benefit from a broader discussion of related AI models(e.g., transformers), to give a better sense of how this work fits into the larger landscape of deep learning applications in medical imaging.
Please see section 4.2 about transformers

Reviewer 4 Report
Comments and Suggestions for Authors The current manuscript uses convolutional neural networks (CNNs) for estimation of fractions of left ventricular ejection upon heart imaging. It is certainly a very important subject; some results are also interesting. A number of changes are, however, necessary prior to acceptance: please address the points listed below. The abstract is not very informative: here the authors should just present their results, without any prose and general-knowledge information. Towards the end of the introduction the authors should concisely but clearly state what they do, how they do it (methods, approaches, models, etc.), and what exactly do they get as the results. A sectioned plan of the entire paper is to be presented here. Image-preparation and analysis of the feature as described on page 4 should be done much more detailed. The current description is too short. Possible biases and feature elimination via pre-treatment of data should be discussed in particular. This is a very important step that deserves proper attention.
In Sec. 2.5 with the statistical analysis the authors are encouraged to mention also other machine-learning methods of image analysis recently introduced in other disciplines. For instance, in Ref. [DOI: https://doi.org/10.1103/PhysRevResearch.5.043129] the physical models of motion of points/tracers on a map were examined and predicted, with some statistical model of diffusion being assessed from the data.
What is the meaning of different colors of points in Fig. 5? The list of literature sources is certainly way too short: the authors should provide an adequate coverage of the subject, from all relevant angles. Particular focus is to be put on the most recent papers. Table 3 is very valuable as a comparison of different approaches: this table can be extended to give the reader a feeling regarding where the results of the current method are most applicable.
Round 2
Reviewer 4 Report
Comments and Suggestions for Authors
The authors should spend more time and deliver a better revision, where all the points raised in the original report would be deeply and fully addressed.
Author Response
In response to your recommendation, which regarding the need for a more comprehensive review of the literature on automatic LVEF determination, we have identified several recent relevant articles previously unknown to us. We intend to incorporate these findings into our discussion and Table 3.
Comment: The list of literature sources is certainly way too short: the authors should provide an adequate coverage of the subject, from all relevant angles. Particular focus is to be put on the most recent papers.
Table 3 is very valuable as a comparison of different approaches: this table can be extended to give the reader a feeling regarding where the results of the current method are most applicable.
Reply: In the discussion, we have provided a much broader reflection on the issue of automatic LVEF by AI by integrating the literature on LV segmentation.
Table 3 has been significantly enriched.
The number of bibliographic references has thus been largely increased.
Kind regards,
Ph Germain
Round 3
Reviewer 4 Report
Comments and Suggestions for Authors
OK revision